# The Dicarboxylate Transporters from the AceTr Family and Dct-02 Oppositely Affect Succinic Acid Production in *S. cerevisiae*

**DOI:** 10.3390/jof8080822

**Published:** 2022-08-06

**Authors:** Toni Rendulić, Frederico Mendonça Bahia, Isabel Soares-Silva, Elke Nevoigt, Margarida Casal

**Affiliations:** 1Centre of Molecular and Environmental Biology (CBMA), Department of Biology, University of Minho, Campus de Gualtar, 4710-057 Braga, Portugal; 2Institute of Science and Innovation for Bio-Sustainability (IB-S), University of Minho, Campus de Gualtar, 4710-057 Braga, Portugal; 3School of Science, Jacobs University Bremen gGmbH, Campus Ring 1, 28759 Bremen, Germany

**Keywords:** Ato1, SatP, Dct-02, succinate, export, transporter, engineering, yeast, anion, channel

## Abstract

Membrane transporters are important targets in metabolic engineering to establish and improve the production of chemicals such as succinic acid from renewable resources by microbial cell factories. We recently provided a *Saccharomyces cerevisiae* strain able to strongly overproduce succinic acid from glycerol and CO_2_ in which the Dct-02 transporter from *Aspergillus niger*, assumed to be an anion channel, was used to export succinic acid from the cells. In a different study, we reported a new group of succinic acid transporters from the AceTr family, which were also described as anion channels. Here, we expressed these transporters in a succinic acid overproducing strain and compared their impact on extracellular succinic acid accumulation with that of the Dct-02 transporter. The results show that the tested transporters of the AceTr family hinder succinic acid accumulation in the extracellular medium at low pH, which is in strong contrast to Dct-02. Data suggests that the AceTr transporters prefer monovalent succinate, whereas Dct-02 prefers divalent succinate anions. In addition, the results provided deeper insights into the characteristics of Dct-02, showing its ability to act as a succinic acid importer (thus being bidirectional) and verifying its capability of exporting malate.

## 1. Introduction

Succinic acid is a platform chemical used in the polymer, food, and cosmetic industries and it is primarily being produced from fossil fuels through chemical synthesis. Caused by the finite fossil fuel reserves and their fluctuating prices, the production of succinic acid by microorganisms has received significant attention from the scientific community and biotech companies as a competitive alternative. Microbial cell factories utilize renewable feedstocks and, as a result, support the global effort at achieving a sustainable economy [1]. Several commercial biotechnological processes have been developed thus far, employing fungal and bacterial hosts, and using glycerol, glucose, and other carbohydrates as substrates [2]. Metabolic and evolutionary engineering strategies have helped to enhance titer, yield, and the productivity of production hosts, and improving succinic acid export has been one important target in these endeavors [2].

The export of succinic acid molecules into the fermentation broth through the plasma membrane is an essential cellular process for efficient and cost-effective succinic acid production via microbial cell factories, as it increases volumetric productivity, relieves the cells from product toxicity, and reduces the feedback inhibition of key enzymes. In addition, it simplifies the downstream processing [1,3,4]. Notably, the intracellularly produced succinic acid cannot pass through the plasma membrane via simple diffusion since virtually all molecules are either fully or partially deprotonated at the near-neutral intracellular pH values of microbial cells and therefore possess a negative charge. As such, succinic acid must be exported by the action of specific plasma membrane transporters [5]. Therefore, successful metabolic engineering approaches included the overexpression of native or heterologous succinic acid exporter genes in host organisms to boost succinic acid production [1].

Among the cellular factories exploited for succinic acid production at commercial scale, the yeast *Saccharomyces cerevisiae* has been attractive due to its robustness at low pH values, allowing convenient downstream processing [6]. The importance of efficient exporters in succinic acid production in baker’s yeast has been demonstrated with the expression of the dicarboxylate transporter Mae1 from *Schizosaccharomyces pombe* [7] and its homologue Dct-02 from *Aspergillus niger* [8]. Mae1 is described as an anion channel which belongs to the Slow-Anion Channel (SLAC1, TC# 2.A.16) family [9]. Dct-02 shares 36.8% protein sequence similarity with Mae1 and it is also annotated as an anion channel belonging to the SLAC1 family, based on protein homology and 3D structure prediction. The expression of Mae1 improved succinic acid production by a factor of five in the industrially patented *S. cerevisiae* SUC-200 strain [10]. This metabolically engineered strain produced 11.9 g/L of succinic acid after 10 days of cultivation in CaCO_3_-buffered shake flasks. The SUC-201 strain, which possesses the same metabolic modifications as the SUC-200 strain but lacks the Mae1, produced only 2.5 g/L of succinic acid [10]. The industrially patented *S. cerevisiae* SUC-632 strain had the *MAE1* expression cassettes replaced by the *DCT-02* expression cassettes, and it was reported to reach succinic acid yields of 0.52 g/g_glucose_ after 70 h of bioreactor cultivation [8].

In our lab, we constructed a *S. cerevisiae* strain (UBR2_CBS_-DHA-SA-AnDCT-02) able to produce succinic acid from glycerol, allowing a higher maximum theoretical yield due to glycerol’s higher degree of reduction compared to glucose [11]. The strain carries an expression cassette for the Dct-02 transporter and the results obtained with an isogenic strain lacking the transporter confirmed the importance of Dct-02 for achieving significant succinic acid production in *S. cerevisiae* (unpublished data). An improved succinic acid producer (UBR2_CBS_-DHA-SA-AnDCT-02 (2)-PYC2oe) has recently been published by Malubhoy, et al. [12]. This strain produced 35 g/L of succinic acid from glycerol after four days of cultivation in CaCO_3_-buffered shake flasks and 20 g/L of succinic acid after three days of cultivation in unbuffered shake flasks, reaching yields of 0.60 and 0.53 g/g_glycerol_, respectively.

The exact mechanism of the succinic acid transport by Dct-02 is not fully understood. Moreover, the question arose whether other transporters from the SLAC1 family or other families exist which might be even more suitable to achieve the goal of high extracellular succinic acid accumulation in media with low pH. In this context, it has been intriguing that several members of the Acetate Uptake Transporter (AceTr, TC# 2.A.96) family and mutated versions thereof have been demonstrated to possess succinic acid transport activity [13,14,15,16,17]. In fact, their effects on succinic acid export in succinic acid overproducing strains have not been evaluated yet. For example, the mutated Ato1^L219A^ transporter from *S. cerevisiae* contains the L219A amino acid substitution in the constrictive site of its pore, which has been shown to enable succinic acid transport [16]. This is in contrast to the wild-type Ato1 which can only transport monocarboxylic acids such as acetic and lactic acids [18]. In addition, we demonstrated that another variant, Ato1^E144A, L219A^, possesses the E144A substitution in its cytosolic loop, which significantly increases its activity and thus turns it into a hyperactive succinic acid transporter in comparison with the Ato1^L219A^.

SatP is a bacterial homologue of Ato1 [13] and it is functional in yeast [16]. The Sat^L131A^ transporter variant contains a substitution in its constrictive site (the L131A) such as Ato1^L219A^, and it was shown to possess a high transport activity for succinic acid similar to the Ato1^E144A, L219A^ variant [16]. Recent structural studies of SatP indicate that members of the AceTr family function as anion channels [19,20], which portrays them as promising candidates for the expression in yeast cell factories engineered for succinic acid production, similarly to the members of SLAC1 family (i.e., Mae1, Dct-02). However, it is unknown whether they mediate the transport of fully deprotonated (divalent) or partially deprotonated (monovalent) succinate anions. This characteristic may affect their applicability as exporters in microbial cell factories because (i) nearly all succinic acid is present in its divalent form at near-neutral pH values inside the cytosol [21] and (ii) the export of the divalent form is more energetically feasible than the export of the monovalent form [22].

In this work, we studied the effects of expressing the transporter variants Ato1^L219A^, Ato1^E144A, L219A^, and SatP^L131A^ in a succinic acid overproducing strain and compared the performance of the resulting strains with an isogenic strain expressing Dct-02 instead. First, we constructed a *S. cerevisiae* SA strain with a very efficient succinic acid production pathway but without a heterologous exporter. The strain construction was based on our previous experience regarding succinic acid production from glycerol [11,12,23]. The resulting SA strain was used as a platform to express and characterize the above transporter variants. We performed unbuffered and CaCO_3_-buffered shake flask cultivations and tracked the effects of transporter expression on succinic acid and by-product formation, glycerol consumption, and biomass formation. Based on the obtained results, we evaluate the applicability of tested transporters in microbial cell factories and further elucidate their mode of action.

## 2. Materials and Methods

### 2.1. Strains, Plasmids and Maintenance

The *S. cerevisiae* strains and plasmids used in this study are listed in Table 1 and Table 2, respectively. Yeast cells were routinely grown on solid YPD medium which contained yeast extract (10 g/L), peptone (20 g/L), glucose (20 g/L), and agar (20 g/L). Agar plates were cultivated in a static incubator at 30 °C. Media were supplemented with hygromycin B (300 mg/L), or nourseothricin (100 mg/L) for selection purposes when needed. *Escherichia coli* DH5α cells were used for plasmid isolation, which were routinely grown in lysogeny broth (LB) containing NaCl (10 g/L), yeast extract (5 g/L), peptone (10 g/L) and adjusted to a pH of 7.5 with 2 M NaOH [24]. For selection and maintenance of plasmid-containing cells, ampicillin (100 mg/L) was added. Cultivations were performed on an orbital shaker at 250 rpm and 37 °C and plasmids were isolated by using the GeneJET^TM^ Plasmid Miniprep Kit (Thermo Fisher Scientific, Waltham, MA, USA).

### 2.2. General Molecular Biology Techniques

Preparative PCRs for the amplification of expression cassettes (or parts thereof) and for sequence determination of integrated expression cassettes were performed using Phusion^®^ High-Fidelity DNA Polymerase (New England BioLabs, Frankfurt am Main, Germany). PCR conditions were adapted to the guidelines of the manufacturer. PCR products were purified by using the GeneJET^TM^ PCR Purification Kit (Thermo Fisher Scientific). Transformation of *S. cerevisiae* with plasmids as well as linear expression cassettes for genomic integration was performed according to the lithium acetate method described by [27].

### 2.3. General Strategies for Genomic Integrations Via CRISPR-Cas9

For genomic integrations via CRISPR-Cas9, the YPRCτ3 on chromosome XVI [28] and position XI-3 [26] were used as integration sites. In order to target Cas9 to the aforementioned genomic locations, the vectors p426-SNR52p-gRNA.YPRCt3-SUP4t-hphMX and pCfB3045 which carry expression cassettes for the respective gRNAs were used (Table 2). According to the employed resistance marker for gRNA expression, Cas9 was expressed using either plasmid p414-TEF1p-Cas9-CYC1t-nat1 or p414-TEF1p-Cas9-CYC1t-hphMX (Table 2). DNA fragments for assembly and integration (either entire expression cassettes or parts thereof) were PCR-amplified from the vectors listed in Table 2 or from genomic DNA isolated from strain CEN.PK113-1A *UBR2_JL1_ GUT1_JL1_* [29]. The used primers (Appendix A) contained 5′-extensions generating 40–60 bp sequences homologous to regions directly upstream and downstream of the inserted double strand break at the integration site or to the respective adjacent fragment (in case several cassettes were assembled at the same locus). Co-transformation of the *S. cerevisiae* strain expressing the Cas9 endonuclease with equimolar amounts of the expression cassettes and the respective vector for gRNA expression resulted in assembly and integration of all expression cassettes at the target locus. Positive transformants were selected on YPD agar containing both nourseothricin and hygromycin B. Both vectors were subsequently removed from the resulting clone by serial transfers in YPD medium lacking the respective antibiotics yielding the desired strain. Subsequently, all integrated expression cassettes were sequenced.

### 2.4. Construction of the SA Strain

The *S. cerevisiae* CEN 2PW strain (Table 1), which was previously constructed by Bahia, et al. [23], served as a starting point of this study. The CEN 2PW strain was of CEN.PK113-1A background, having its endogenous *UBR2* and *GUT1* alleles replaced by those from CBS 6412-13A and CEN.PK113-7D *JL1*, respectively [29]. Moreover, the CEN 2PW possessed the expression cassettes for *Opgdh* from *Ogataea parapolymorpha* (under the control of *TEF1* promoter and *CYC1* terminator), *DAK1* (under the control of *ADH2* promoter and *TPS1* terminator), and *CjFPS1* from *Cyberlindnera jadinii* (under the control of *TDH3* promoter and *RPL15A* terminator), which were integrated into the YGLCτ3 locus [23]. For the establishment of succinic acid overproducing rTCA pathway, the expression cassettes for *ScMDH3-R* (under the control of *JEN1* promoter and *IDP1* terminator), *RofumR* (under the control of *HOR7* promoter and *DIT1* terminator), and *TbFRDg-R* (under the control of *FBA1* promoter and *ADH1* terminator) were amplified from the plasmids pUC18-P*_JEN1_-MDH3-R*, pUC18-P*_HOR7_-RofumR*, and pUC18-P*_FBA1_-TbFRDg-R* (Table 2) using the primer pairs listed in Appendix A. These cassettes were subsequently assembled and integrated at the *YPRCτ3* locus of strain CEN 2PW by employing the CRISPR-Cas9 system as described above to obtain the SA strain.

### 2.5. Construction of SA-Dct-02, SA-Ato1 (S), SA-Ato1 (HS), SA-SatP (HS), and SA-_COX7_SatP (HS) Strains

The cassettes for expression of Dct-02, Ato1^L219A^, Ato1^E144A, L219A^, SatP^L131A^ (under the control of the *TDH3* promoter and the *CYC1* terminator) were assembled and integrated using CRISPR-Cas9 at position XI-3 in the genome of the SA strain, resulting in the creation of SA-Dct-02, SA-Ato1 (S), SA-Ato1 (HS), and SA-SatP (HS) strains, respectively (Table 1). Additionally, the cassette for expression of SatP^L131A^ under the control of the *COX7* promoter and the *CYC1* terminator was assembled and integrated in the SA strain by applying the same strategy, resulting in the construction of the SA-_COX7_SatP (HS) strain. Transporter coding sequences were amplified from the plasmids listed in Table 2, while promoter and terminator sequences were amplified from genomic DNA of strain CEN.PK113-1A UBR2_JL1_ GUT1_JL1_, using the primers listed in (Appendix A).

### 2.6. Isolation of Genomic DNA from S. cerevisiae Transformants and Diagnostic PCR

Correct integration of all expression cassettes was verified by diagnostic PCR using OneTaq Quick-load DNA polymerase and buffer according to the manufacturer’s guidelines (New England Biolabs). Genomic DNA was isolated according to a modified protocol from Hoffman and Winston [30]. Approximately 50 mg of cells were suspended in 200 µL of TE buffer (10 mM Tris, 1 mM EDTA, pH 8.0). Subsequently, 300 mg of acid-washed glass beads (diameter of 0.425–0.6 mm) and 200 µL of phenol:chloroform:isoamyl alcohol (25:24:1) were added. The tubes were vortexed at maximum speed for 2 min and centrifuged at 15,700 g for 10 min. The aqueous phase (1 µL) was used as a template in 25 µL PCR reactions. PCR primers were designed to bind upstream and downstream of the genomic integration sites. For analyzing integrations of multiple expression cassettes, additional primers were designed to produce amplicons covering the junctions between the individual integrated expression cassettes.

### 2.7. Media and Cultivation Conditions for the Production of Succinic Acid from Glycerol

All pre- and intermediate cultures were cultured in synthetic medium-containing glucose (20 g/L) and ammonium sulfate (5 g/L) as the carbon and nitrogen source, respectively. All experiments for assessing succinic acid production in shake flask batch cultivation were performed in synthetic medium-containing 60 mL/L (75.6 g/L) glycerol as the sole carbon source with urea as the nitrogen source (2.8 g/L). The synthetic medium was prepared according to Verduyn, et al. [31] containing 3 g/L KH_2_PO_4_, 0.5 g/L MgSO_4_·7H_2_O, 15 mg/L EDTA, 4.5 mg/L ZnSO_4_·7H_2_O, 0.84 mg/L MnCl_2_·2H_2_O, 0.3 mg/L CoCl_2_·6H_2_O, 0.3 mg/L CuSO_4_·5H_2_O, 0.4 mg/L NaMoO_4_·2H_2_O, 4.5 mg/L CaCl_2_·2H_2_O, 3 mg/L FeSO_4_·7H_2_O, 1 mg/L H_3_BO_3_, and 0.1 mg/L KI. After heat sterilization of the medium, filter sterilized vitamins were added. The final vitamin concentrations were: 0.05 mg/L D- (+) -biotin, 1 mg/L D-pantothenic acid hemi-calcium salt, 1 mg/L nicotinic acid, 25 mg/L myo-inositol, 1 mg/L thiamine chloride hydrochloride, 1 mg/L pyridoxine hydrochloride, and 0.2 mg/L 4-aminobenzoic acid. In case urea was used as the nitrogen source (in main culture media), an appropriate aliquot of a stock solution was added after autoclaving to obtain a final concentration of 2.8 g/L. The pH of the synthetic glucose medium was adjusted to 6.5 with 4 M KOH. The pH of unbuffered synthetic glycerol medium was adjusted to 4.0 with 2 M H_3_PO_4_, while the pH of CaCO_3_-buffered medium was adjusted to 6.0 or 7.0 with KOH, before CaCO_3_ addition. The 100 mM potassium-phosphate-buffered synthetic glycerol medium with a pH of 7.5 was prepared by adding 12.8 g/L K_2_HPO_4_ and 3.6 g/L KH_2_PO_4_ (instead of the standard 3.0 g/L KH_2_PO_4_).

For pre-cultivation, cells from a single colony were used to inoculate 3 mL of the synthetic glucose medium in a 10 mL glass tube and incubated at an orbital shaking of 200 rpm and 30 °C overnight. The pre-culture was used to inoculate 10 mL of the same medium in a 100 mL Erlenmeyer flask (closed with a metal cap) adjusting an OD_600_ of 0.2. This culture, hereafter referred to as intermediate culture, was cultivated at the same conditions for 24 h. The appropriate culture volume from the intermediate culture (in order to later adjust an OD_600_ of 0.2 in 100 mL of synthetic glycerol medium) was centrifuged at 800 g for 5 min and the supernatant was discarded. The cell pellet was then washed once by re-suspending the cells in a synthetic glycerol medium. The cell suspension was centrifuged again and re-suspended in 100 mL of the same medium in a 500 mL Erlenmeyer flask (closed with a cotton plug), adjusting to a final OD_600_ of 0.2. The main cultures were incubated at orbital shaking of 200 rpm and 30 °C and samples for OD_600_ determination and HPLC analysis were taken at regular time intervals. For those cultures which were supplemented with CaCO_3_, 3 g or 10 g of CaCO_3_ (in order to achieve 30 g/L or 100 g/L final CaCO_3_ concentration, respectively) were transferred to a 500 mL shake flask and autoclaved. Main cultures (100 mL) with an initial OD_600_ of 0.2 were prepared as described above and subsequently the entire cell suspensions were added to the shake flasks containing the CaCO_3_ under sterile conditions. For OD_600_ measurements, samples were diluted in 0.2 M HCl ensuring the complete dissolving of the suspended CaCO_3_.

### 2.8. Metabolite Analysis by HPLC

Samples of culture supernatants (1 mL) were first filtered through 0.2 mm Minisart RC membrane filters (Sartorius, Göttingen, Germany) and, if required, were stored at −20 °C until analysis. The concentrations of succinic acid, malic acid, acetic acid, glycerol, and ethanol in culture media were determined using a Waters HPLC system (Eschborn, Germany) consisting of a binary pump system (Waters 1525), injector system (Waters 2707), the Waters column heater module WAT038040, a refractive index (RI) detector (Waters 2414), and a dual wavelength absorbance detector (Waters 2487). The samples were loaded on an Aminex HPX-87H cation exchange column (Bio-Rad, München, Germany) coupled to a Micro-guard column (Bio-Rad) and eluted with 5 mM H_2_SO_4_ as the mobile phase at a flow rate of 0.6 mL/min and a column temperature of 45 °C. Volumes of 20 µL of sample were used for injection. Succinic acid, malic acid, and acetic acid were detected using the dual wavelength absorbance detector (Waters 2487), while glycerol and ethanol were analyzed with the RI detector (Waters 2414). The retention time for malic acid was 9.6 min, for succinic acid it was 11.9 min, for glycerol it was 13.5 min, for acetic acid it was 15.2 min, and for ethanol it was 21.6 min. Data were processed and analyzed using the Breeze 2 software (Waters).

## 3. Results

### 3.1. Construction of the Baseline SA Strain

To evaluate the impact of the alternative transporter variants Ato1^L219A^, Ato1^E144A, L219A^, and SatP^L131A^ on succinic acid production, in comparison to the previously used Dct-02, we required a suitable baseline strain. For this purpose, we constructed the *Saccharomyces cerevisiae* strain engineered for succinic acid overproduction from glycerol as described in Materials and Methods and named it ‘strain SA’ throughout this study (Table 1). In summary, the strain SA (Figure 1) was equipped with the L-G3P and DHA pathways for efficient glycerol utilization [23], the *UBR2* allele from the strain CBS 6412-13A, which is required for CEN.PK strains to grow in synthetic glycerol medium [32], and the reductive TCA (rTCA) pathway [12] for efficient cytosolic succinic acid production.

The engineered SA strain was able to efficiently utilize glycerol as a sole carbon source, both in unbuffered and CaCO_3_-buffered shake flask cultivations, reaching maximum OD_600_ values of 27.2 and 30.5, respectively (Figure 2). The SA strain produced up to 5.6 g/L of succinic acid during unbuffered shake flask cultivation, and up to 13.9 g/L of succinic acid during CaCO_3_-buffered shake flask cultivation (Figure 2), despite lacking a heterologous succinic acid transporter. Maximum observed succinic acid yields were 0.076 ± 0.0010 g/g_glycerol_ (at 144 h) and 0.182 ± 0.0058 g/g_glycerol_ (at 288 h), respectively. Upon glycerol exhaustion in the unbuffered medium, the strain SA reconsumed 4.0 g/L of succinic acid from the media, whereas no succinic acid reconsumption was observed in the CaCO_3_-buffered medium (Figure 2). The observed succinic acid reconsumption only occurred when the pH values of the medium were between 3.5 and 3.8, i.e., below the pKa_1_ of succinic acid (4.2). Therefore, this phenomenon could be attributed to simple diffusion of uncharged succinic acid molecules into the cells at low external pH. Ethanol was the major by-product of the SA strain metabolism, reaching titers of up to 10.3 and 10.4 g/L in unbuffered and CaCO_3_-buffered media, respectively (Figure 2). Notably, ethanol is also the major by-product of an isogenic strain (CEN 2PW) which possesses the L-G3P and DHA pathways for glycerol catabolism but lacks the modifications required for succinic acid overproduction (engineered rTCA pathway) [23]. After glycerol exhaustion, nearly all ethanol was re-consumed by the strain SA in both conditions (Figure 2). Notably, the strain SA did not produce any malic acid (Figure 2), which is an intermediate of the rTCA pathway (Figure 1), and a common by-product of yeast cell factories engineered for succinic acid production [12,33,34].

### 3.2. Impact of Dct-02 Expression on Extracellular Succinic Acid Accumulation in the Baseline Strain SA

To study the impact of Dct-02 on the time course of extracellular succinic acid concentration, we integrated the respective expression cassette (coding sequence flanked by *TDH3* promoter and *CYC1* terminator) into the genome of the strain SA which resulted in the strain SA-Dct-02 (Table 1). The expression of Dct-02 negatively affected the growth (Figure 2). Indeed, the maximal biomass accumulated by strain SA-Dct-02 throughout the cultivation was only 70.1% that of strain SA in unbuffered medium and 58.1% in CaCO_3_-buffered medium. Still, it was able to utilize all glycerol from the medium in both conditions. As expected, the expression of Dct-02 significantly improved the succinic acid production in terms of obtained maximum titers and yields. The strain SA-Dct-02 produced up to 26.2 g/L of succinic acid in unbuffered medium, and 29.3 g/L of succinic acid in CaCO_3_-buffered medium (Figure 2). This corresponds to 4.7-fold and 2.1-fold improvement compared to the strain SA, respectively. Maximum observed succinic acid yields were 0.418 ± 0.0051 g/g_glycerol_ (at 96 h) and 0.419 ± 0.0178 g/g_glycerol_ (at 144 h), corresponding to 5.5-fold improvement in unbuffered medium and 2.3-fold improvement in CaCO_3_-buffered medium, respectively, in comparison with the strain SA.

Interestingly, the expression of Dct-02 led to succinic acid import in the later phase of the CaCO_3_-buffered cultivation. Indeed, the strain SA-Dct-02 re-consumed 19.4 g/L of succinic acid upon glycerol exhaustion and the final succinic acid titer (at 288 h) was lower than the one observed for the SA strain (Figure 2). Data imply that the import occurred in a phase when the pH values in the medium were too high (pH 5.0–6.4) to support simple diffusion of succinic acid. In the unbuffered medium, the expression of Dct-02 did not significantly contribute to succinic acid import since the SA-Dct-02 re-consumed only 5.2 g/L of succinic acid upon glycerol exhaustion, similarly to the SA strain, which could be attributed to simple diffusion as it occurred at pH 3.2.

In contrast to the strain SA, malic acid was the major by-product of the strain SA-Dct-02 when cultivated in CaCO_3_-buffered medium, showing that Dct-02 functioned as an efficient malic acid exporter (Figure 2). Overall, the observed malic acid export could be divided into two distinct phases. During the phase of glycerol consumption, malic acid was exported alongside succinic acid as an intermediate of the same pathway. During the phase of glycerol exhaustion, succinic acid seemed to be imported while similar amounts of malic acid were concurrently exported out of the cells, indicating reutilization of succinic acid as carbon and energy source via its conversion into malic acid in the cytosol or mitochondria. In the unbuffered medium, the detected extracellular malic acid concentrations were significantly lower (Figure 2). The expression of Dct-02 led to a significant reduction of ethanol formation as it instead supported the production of succinic acid and malic acid (Figure 2). In conclusion, Dct-02 improved succinic acid production in the SA strain in both conditions with a higher overall impact in unbuffered cultivation medium, and behaved as a bidirectional succinic acid and malic acid transporter.

### 3.3. Impact of Engineered AceTr Homologues on Succinic Acid Production of the Strain SA

To study the impact of Ato1^L219A^, Ato1^E144A, L219A^, and SatP^L131A^ on succinic acid production, we integrated the respective expression cassettes (coding sequence flanked by *TDH3* promoter and *CYC1* terminator) into the genome of the strain SA. The resulting strains were named SA-Ato1 (S), SA-Ato1 (HS), and SA-SatP (HS), respectively (Table 1). These strains were first characterized in shake-flask cultivations in the unbuffered synthetic glycerol medium. The expression of Ato1^L219A^, Ato1^E144A, L219A^, and SatP^L131A^ slightly enhanced the growth during the early phase of cultivation when compared to the strain SA. However, the final biomass values reached at the end of the cultivation were lower in the strains expressing these transporters (Figure 3). The described negative impact on growth was most prominent in the strain SA-SatP (HS), followed by the SA-Ato1 (HS) strain, and less pronounced in the SA-Ato1 (S) strain (Figure 3). Overall, the expression of the engineered AceTr homologues did not impair the cell growth as much as Dct-02 (Figure 2). Interestingly, the expression of Ato1^L219A^, Ato1^E144A, L219A^, and SatP^L131A^ reduced succinic acid production in comparison to the strain SA, which is in strong contrast to the positive effect described above for the expression of Dct-02 in this strain (Figure 2). These data indicate that the AceTr homologues favored succinic acid import even during the phase of glycerol consumption.

Among the strains which expressed the AceTr homologues, the SA-Ato1 (S) strain was the most similar to the SA strain with respect to the initial kinetics of extracellular succinic acid accumulation, and a negative impact of Ato1^L219A^ expression could not be observed before 96 h of cultivation. The maximum concentration of succinic acid detected for the SA-Ato1 (S) strain was 3.9 g/L, i.e., a 30.4% decrease in comparison with the SA strain. The maximum observed succinic acid yield was 0.070 ± 0.0009 g/g_glycerol_ (at 96 h), which represents a slight decrease (7.6%) compared to the SA strain. Moreover, the expression of Ato1^L219A^ caused an early onset of succinic acid re-consumption when there were still 13.9 g/L of glycerol left in the medium (Figure 3). Compared to Ato1^L219A^, the expression of Ato1^E144A, L219A^ and SatP^L131A^ resulted in different kinetics of extracellular succinic acid concentration. In fact, the SA-Ato1 (HS) and SA-SatP (HS) strains displayed hindered succinic acid export already at the very beginning of cultivation (Figure 3). The observed behavior also had an effect on measured external pH, which was higher for these two strains throughout the cultivation, in comparison with strains SA-Ato1 (HS) and SA. The Ato1^E144A, L219A^ expression reduced the accumulation of extracellular succinic acid more severely than the Ato1^L219A^, as the strain Ato1 (HS) produced only up to 2.5 g/L of succinic acid (a 55.4% decrease in comparison with the strain SA). The maximum observed succinic acid yield of the strain Ato1 (HS) was only 0.034 ± 0.0013 g/g_glycerol_ (at 72 h), which is a 55.3% decrease in comparison with the strain SA. The expression of SatP^L131A^ affected extracellular succinic acid accumulation more negatively than Ato1^E144A, L219A^ in the initial phase of the cultivation, however, the strain SA-SatP (HS) accumulated significantly more succinic acid in the medium compared to the strain SA-Ato1 (HS) at the end of the cultivation, reaching a maximum succinic acid titer of 4.8 g/L and a yield of 0.061 ± 0.0056 g/g_glycerol_ (at 240 h). Nonetheless, these values represent 14.3% and 19.7% decreases in maximum obtained succinic acid titers and yields, respectively, in comparison with the strain SA.

Interestingly, the expression of engineered AceTr homologues led to distinct phenotypes in terms of ethanol production. The strain SA-Ato1 (S) produced ethanol in a similar amount compared to the strain SA (Figure 3). However, the strain SA-Ato1 (HS) produced more ethanol (18.7% increase), while the strain SA-SatP (HS) produced less ethanol (19.7% decrease) than the strain SA (Figure 3). Similarly to the strain SA, strains expressing engineered AceTr homologues did not secrete any acetic or malic acid in the unbuffered medium, as these acids were not detected via HPLC analysis. In conclusion, these experiments showed that the tested engineered AceTr homologues, particularly Ato1^E144A, L219A^ and SatP^L131A^, rather acted as succinic acid importers at low external pH, thus hindering a net export of succinic acid compared to the baseline SA strain.

### 3.4. External pH Modulated the Activity of Engineered AceTr Homologues

Taymaz-Nikerel, et al. [21] showed that higher external pH increases the energetic feasibility of succinic export by plasma membrane transporters. In this regard, transporters which mediate succinic acid import under the conditions of low external pH (e.g., pH 3.0), may operate in the reverse direction at higher external pH values (e.g., pH 7.0), thus functioning as exporters [21]. To test whether higher external pH could alter the impact of the engineered AceTr homologues on succinic acid production, we performed a CaCO_3_-buffered shake flask cultivation of the strains Sa-Ato1 (S), SA-Ato1 (HS), SA-SatP (S), and SA. Surprisingly, the expression of Ato1^E144A, L219A^ and SatP^L131A^ considerably extended the lag phase under these conditions. In fact, these strains entered the exponential phase of growth only four to five days after the strains SA-Ato1 (S) and SA (Figure 4). Notably, the strains Ato1 (HS) and SatP (HS) did not display such growth defects when cultivated in the unbuffered medium (Figure 3).

Although the strains SA-Ato1 (HS) and SA-SatP (HS) showed this delayed onset of growth, all tested strains reached similar maximum succinic acid titers (~13.5 g/L) and yields (~0.18 g/g_glycerol_). In addition, no reconsumption of succinic acid was observed in any of the tested strains, indicating that Ato1^L219A^, Ato1^E144A, L219A^, and SatP^L131A^ had an altered transport activity in the presence of CaCO_3_ and at higher external pH.

Another interesting phenomenon was the increased acetic acid secretion by the strains SA-Ato1 (HS) and SA-SatP (HS) in the CaCO_3_-buffered cultivations, indicating that the Ato1^E144A, L219A^ and SatP^L131A^ promoted acetic acid export (Figure 4), which is in strong contrast to the behavior of the Dct-02 transporter under the same conditions, as no acetic acid was detected in the cultivation medium of the SA-Dct-02 strain. The strain SA-Ato1 (S) produced similar amounts of ethanol compared to the strain SA, whereas the strains SA-Ato1 (HS) and SA-SatP (HS) produced less (Figure 4). No malic acid secretion was observed in any of the tested strains. In conclusion, the presence of the CaCO_3_ and/or the higher pH removed the diminishing effects of the tested AceTr homologues on extracellular succinic acid accumulation. Additionally, the two hyperactive AceTr homologues increased acetic acid export and caused a prolonged lag phase of growth under these conditions.

### 3.5. Understanding the Prolonged Lag Phase of the Strains SA-Ato1 (HS) and SA-SatP (HS) in CaCO_3_-Buffered Shake Flask Cultivations

We were intrigued by the considerable lag phases of the strains SA-Ato1 (HS) and SA-SatP(HS) during CaCO_3_-buffered shake flask cultivations, and we considered the various possible reasons. Our initial hypothesis was that the strong overexpression caused a burden for the cells. We therefore tested the *COX7* promoter to control the expression of SatP^L131A^. The latter is known to be weaker than the *TDH3* promoter in synthetic glycerol medium [35]. In order to also check the effect of the CaCO_3_ concentration, the strain SA-_COX7_SatP (HS) (Table 1) and the baseline strain SA were cultivated in shake flasks with 30 g/L (standard condition) and 100 g/L CaCO_3_. Despite the weaker promoter, the SA-_COX7_SatP (HS) still displayed a prolonged lag phase in comparison with the strain SA (Figure 5A). Moreover, the duration of the lag phase of the strain SA-_COX7_SatP (HS) was higher when the increased CaCO_3_ concentration was used (keeping the pH at higher levels throughout the cultivation), whereas the strain SA behaved the same in both conditions (Figure 5A). To check whether the lag phase is directly caused by the presence of CaCO_3_, or rather due to the higher external pH (which is a consequence of the basic properties of CaCO_3_), we also cultivated the strains SA-Ato1 (HS), SA-SatP (S), and SA in a potassium-phosphate-buffered (100 mM) synthetic glycerol medium adjusted to pH 7.5. The strains SA-Ato1 (HS) and SA-SatP (S) also displayed their long lag phases in this medium (Figure 5B). Therefore, our data suggest that the Ato1^E144A, L219A^ and SatP^L131A^ expression made the SA strain sensitive to high external pH.

## 4. Discussion

### 4.1. Succinic Acid Export by the SA Strain

The *S. cerevisiae* baseline strain used in this study (strain SA) was primarily engineered to accumulate succinic acid intracellularly as it did not express any known succinic acid exporters. Nonetheless, the SA strain was still able to produce up to 5.6 and 13.9 g/L of succinic acid in unbuffered and CaCO_3_-buffered shake flasks cultivations with maximum yields of 0.08 and 0.18 g/g_glycerol_, respectively. Notably, the parental *S. cerevisiae* strain possessing the same catabolic pathways for glycerol utilization but lacking the rTCA pathway for succinic acid overproduction did not secrete any succinic acid in similar conditions [23], showing that the overexpression of the rTCA pathway triggered the observed export of succinic acid by the SA strain. The native succinate export was also observed in the engineered *S. cerevisiae* strain PMCFfg constructed by Yan et al. (2014), which was equipped with the rTCA pathway for succinic acid overproduction and lacked a heterologous succinic acid exporter, like the strain SA. The strain PMCFfg reached a succinic acid titer of 13.0 g/L and yield of ~0.14 g/g_glucose_ in a CO_2_-sparged bioreactor cultivation with glucose as the sole carbon source [36].

As no native specific succinic acid transporter was identified so far in *S. cerevisiae*, this species most probably possesses an as yet unidentified specific succinic acid exporter(s) in its plasma membrane, or other exporter(s) (e.g., of small solutes) became leaky to succinic acid molecules within the SA strain. This/these hypothetical transporter(s) exported succinic acid in all conditions tested, and unlike Dct-02, it/they did not export any malic acid as a by-product, based on the cultivation data of the SA and SA-Dct-02 strains (Figure 2). The identification, overexpression, and further engineering of this/these exporter(s) may be of interest for industrial succinic acid production and will be a focus of our future work.

### 4.2. New Insights into the Effects of Dct-02 Expression in Yeast Cell Factories

Dct-02 from *A. niger* was already employed in the past as a heterologous succinic acid exporter in yeast cell factories which achieved high succinic acid titers and yields [8,11,12]. However, there was little information available about the individual contribution of Dct-02 towards the reported achievements. Our current study directly compared the strains SA and SA-Dct-02 in two distinct cultivation conditions, thus providing more insights into the effects of Dct-02 expression on strain performance. Dct-02 significantly improved succinic acid export even at low pH (Figure 2), thus justifying its usage as an efficient exporter in the industrial strains. Recent findings about the activation of Dct-02 homologue AtSLAC1 from *Arabidopsis thaliana* via posttranslational modification [37] suggest that the export activity of Dct-02 could be improved by introducing phosphomimetic substitutions in its putative phosphorylation sites. Besides exporting succinic acid, Dct-02 also exported significant amounts of malic acid (Figure 2). This was especially the case in the presence of CaCO_3_, which promotes the accumulation of rTCA intermediates by supplying bicarbonate for pyruvate carboxylation (Figure 1). Notably, malic acid export by Dct-02 has previously been reported by Shah, et al. [38]. Moreover, Mae1 from *S. pombe* [39] and several other Dct-02 homologues [40,41,42] were also shown to export malic acid. From the perspective of industrial succinic acid production via yeast cell factories, malic acid represents an unwanted by-product. Therefore, it may be of interest to eliminate the affinity of Dct-02 towards malic acid via transporter engineering. In this study, Dct-02 had a negative effect on growth as visible by comparing the strain SA-Dct-02 with the strain SA. Notably, the conversion of glycerol into succinic acid via the envisaged rTCA pathway does not lead towards net ATP formation, whereas this is not the case for the respiratory dissimilation of glycerol or its conversion into ethanol (Figure 1). On top of that, the export of negatively charged succinic acid anions via anion channels is an electrogenic process, and in order to retain charge homeostasis, cells must concurrently export positively charged protons via proton pumps, whereas expulsion of each proton costs 1 mol ATP [6,21,22]. This means that the export of one divalent succinate anion via an anion channel in principle costs 2 mol additional ATP. Although Dct-02 was so far primarily regarded as an exporter, we showed here that it could also import succinic acid (Figure 2), like its homologue Mae1 from *S. pombe* [43]. The import of succinic acid was observed at external pH values between 5.0 and 6.4 (Figure 2), i.e., when monovalent and divalent succinate anions predominated (pK_a1_ and pK_a2_ values of succinic acid correspond to 4.2 and 5.6, respectively). The observed bidirectional transport by Dct-02 and its preference for negatively charged succinate anions support our claims (see introduction) and the ones by Shah, et al. [38] that Dct-02 functions as an anion channel (uniporter).

### 4.3. Influence of External PH on Succinic Acid Production by the Strains Expressing Ato1^L219A^, Ato1^E144A, L219A^, and SatP^L131A^

Obviously, Ato1^L219A^, Ato^E144A, L219A^ and SatP^L131A^ hindered the extracellular succinic acid accumulation in the unbuffered cultivation medium since the SA-Ato1 (S), SA-Ato1 (HS), and SA-SatP (HS) strains achieved even lower succinic acid titers and yields than the baseline SA strain (Figure 3). Thus, our results indicate that these AceTr homologues favored succinic acid import instead of export in the conditions of low external pH between 3.5 and 5.5. As a net result, the succinic acid exported by the endogenous transporter(s) is assumed to be transferred back to the cells in a futile cycle which is supposed to cost extra energy. One might hypothesize that the observed phenomenon could also occur if the AceTr transporters promoted the export of other carboxylic acids, thereby pulling the carbon flux into the direction of other, rather unwanted, products. However, this hypothesis is unlikely because no other alternative products were detected by HPLC analysis.

The observed distinctions in the initial kinetics of extracellular succinic acid accumulation in the unbuffered medium between the strains expressing different AceTr homologues may be explained by the fact that Ato1^E144A, L219A^ and SatP^L131A^ are hyperactive succinic acid transporters in comparison with Ato1^L219A^ (see introduction). Moreover, in a previous study, Ato1^E144A^ and SatP displayed higher affinity (lower K_m_ values) for acetic acid compared to wild-type Ato1 [16]. Accordingly, Ato1^E144A, L219A^ and SatP^L131A^ most probably also possess higher affinity for succinic acid compared to Ato1^L219A^. On the other hand, at an external pH between 6.0 and 7.5, Ato1^L219A^, Ato^E144A, L219A^ and SatP^L131A^ did not mediate the import or export of succinic acid, since the expression of these transporters resulted in equal amounts of extracellular succinic acid compared to the SA strain by the end of the cultivation (Figure 4). Notably, the strains expressing these transporters did not display succinic acid import even upon glycerol exhaustion. According to the pK_a_ values of succinic acid, the divalent succinate anions predominated in the medium in these conditions. Therefore, the observed influence of the external pH on the behavior of Ato1^L219A^, Ato^E144A, L219A^ and SatP^L131A^ suggests that these transporters prefer the monovalent rather than divalent form of succinate as substrate. One might argue that the AceTr homologues were simply inactive at pH > 6; however, such a hypothesis can be excluded based on the observed significant acetic acid export by Ato^E144A, L219A^ and SatP^L131A^ under these conditions.

The expression of Ato1^L219A^ did not lead to any changes in the phenotype at external pH > 6 compared to cells of the strain SA, whereas the expression of Ato^E144A, L219A^ and SatP^L131A^ caused a significant prolongation of the lag phase (Figure 4 and Figure 5). The prolonged lag phase appears to correlate with the significantly higher production of acetic acid by the respective strains (Figure 4). Notably, Ato^E144A, L219A^ and SatP^L131A^ were previously described as hyperactive acetic acid transporters, whereas Ato1^L219A^ displayed similar acetic acid transport activity as the wild-type Ato1 [16]. No acetic acid production was registered in the conditions of low external pH, and the strains expressing Ato^E144A, L219A^ and SatP^L131A^ did not display prolonged lag phase in these conditions (Figure 3). Acetic acid is a precursor of cytosolic acetyl-CoA, which plays an important role in growth-related processes such as lipid biosynthesis [44]. The elevated export of acetic acid by Ato^E144A, L219A^ and SatP^L131A^ might have reduced the synthesis of acetyl-CoA and thus delayed the growth of the respective strains. According to the thermodynamic analysis of monocarboxylic acid transport by van Maris, et al. [45], higher external pH increases the feasibility of acetic acid export by *S. cerevisiae*. Therefore, the high external pH of the cultivation medium may have promoted the export of acetic acid by the hyperactive homologues Ato^E144A, L219A^ and SatP^L131A^, which in turn may have caused the prolonged lag phases of the respective strains. As the external pH of the medium gradually decreased over time, the respective strains stopped exporting acetic acid and were thus able to exit the lag phase and proceed with unconstrained growth (Figure 4 and Figure 5).

Overall, this study showed that Ato1^L219A^, Ato^E144A, L219A^ and SatP^L131A^ are not suitable as succinic acid exporters in yeast cell factories. In this regard, they behaved similarly to PkJen2-1 and PkJen2-2 from *Pichia kudriavzevii*, which were also demonstrated to hinder succinic acid production [46]. On the other hand, the members of the AceTr family could most probably be utilized as exporters of acetic acid, as suggested by the results of this work, or lactic acid, as suggested by Turner, et al. [47], as long as the cultivation medium would provide sufficiently high external pH.

### 4.4. Why Do Dct-02 and AceTr Members Display Opposite Effects on Succinic Acid Production?

According to our results, Dct-02 mediated succinic acid export, whereas the engineered AceTr members mediated succinic acid import during the same stage of cultivation in the unbuffered medium, i.e., when glycerol was still consumed by the respective strains (Figure 2 and Figure 3). These results demonstrate that the two distinct families of transporters (AceTr and SLAC1) display different properties, as the direction of any transport reaction is determined by its energetics. Taymaz-Nikerel, et al. [21] analyzed the energetics of succinic acid transport, showing that the mechanism of transport determines whether a certain transporter will mediate the import or export of succinic acid, at a defined external pH and at a particular concentration gradient of the substrate. In this regard, the export of divalent succinate anions via anion channels is energetically more favorable than the export of monovalent succinate anions [22]. Therefore, if Dct-02 and the engineered AceTr family transporters all function as anion channels (uniporters, thus electrogenic), then Dct-02 mediates the transport of divalent succinate anions whereas Ato^L219A^, Ato1^E144A, L219A^, and SatP^L131A^ mediate the transport of monovalent anions. These claims are additionally supported by the external pH values at which the respective transporters could mediate succinic acid import, since the engineered AceTr members preferred lower pH values in comparison with Dct-02 (Figure 2, Figure 3 and Figure 4), when there were hardly any divalent succinate anions present in the medium according to the pK_a_ values of succinic acid. Moreover, data obtained by Shah et al. [38] also indicate that Dct-02 mediates the uniport of divalent succinate anions.

## 5. Conclusions

This study reports the impact of dicarboxylate anion channels from two distinct families (AceTr and SLAC1) on extracellular succinic acid accumulation in an *S. cerevisiae* strain engineered for production of succinic acid from glycerol and CO_2_. Depending on the cultivation conditions used in this study, the respective transporters show distinct behaviors (Figure 6).

The expression of AceTr homologues negatively affects succinic acid production at low external pH because they favor succinic acid import rather than export (Figure 6A). This behavior is in strong contrast to Dct-02, which positively affects succinic acid production by efficiently exporting it (Figure 6A). In buffered media with a near-neutral external pH, the AceTr homologues have a neutral effect on succinic acid production, whereas Dct-02 improves it (Figure 6B). In the conditions of near-neutral external pH, Ato1^E144A, L219A^ and SatP^L131A^ expression causes a prolonged lag phase of *S. cerevisiae*, which appears to correlate with the elevated acetic acid export mediated by these transporters (Figure 6B). Additionally, we provide evidence that Dct-02 can export malic acid (Figure 6B–D) and functions as a bidirectional succinic acid transporter (Figure 6D). Altogether, these data indicate that AceTr homologues transport monovalent succinate anions, whereas Dct-02 transports divalent succinate anions. Thus, our study deepened the knowledge on the two groups of succinic acid transporters, demonstrating in vivo how their distinct preferences for monovalent and divalent succinate anions had a decisive impact on the performance of yeast cell factories.

## Figures and Tables

**Figure 1 jof-08-00822-f001:**
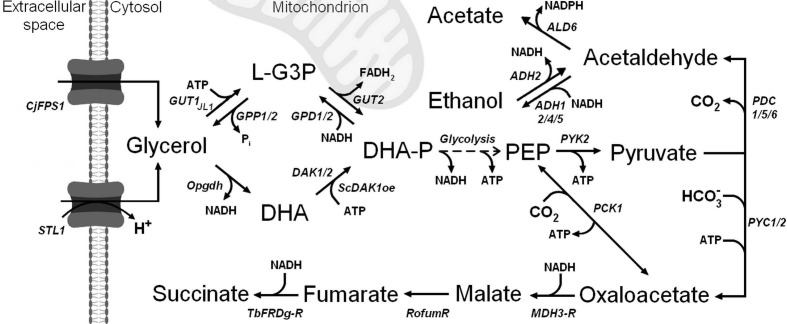
Overview of the central metabolism of the succinic acid overproducing *Saccharomyces cerevisiae* SA strain constructed in this study. This strain was used to study the impact of different transporters from the AceTr family in comparison to the Dct-02 transporter from *A. niger* on succinic acid transport. The strain was designed for the utilization of glycerol as the sole source of carbon. It does not contain any heterologous succinic acid transporters and shows a maximum extracellular succinic acid concentration of 13.9 g/L in shake flask cultivations (see main text). *CjFPS1*: *Cyberlindnera jadinii FPS1* (aquaglyceroporin), *STL1*: glycerol/H^+^ symporter, *GUT1*: glycerol kinase, *GUT2*: FAD-dependent glycerol-3-phosphate dehydrogenase, *GPP1/2*: glycerol-3-phosphate phosphatase, *GPD1/2*: glycerol 3-phosphate dehydrogenase, *Opgdh*: NAD^+^-dependent glycerol dehydrogenase from *Ogataea parapolymorpha*, *ScDAK1oe*: overexpression of *S. cerevisiae* dihydroxyacetone kinase, DAK1/2: dihydroxyacetone kinase, *PYK2*: pyruvate kinase, *PCK1*: phosphoenolpyruvate carboxykinase, *PYC1/2*: pyruvate carboxylase, *MDH3-R*: peroxisomal malate dehydrogenase targeted to the cytosol, *RofumR*: fumarase from *Rhizopus oryzae*, *TbFRD-R*: glycosomal fumarate reductase from *Trypanosoma brucei* retargeted to the cytosol. *PDC1/5/6*: pyruvate decarboxylase, *ADH1/2/4/5*: alcohol dehydrogenase, *ALD6*: aldehyde dehydrogenase, L-G3P: L-glycerol-3-phosphate, DHA: dihydroxyacetone, DHA-P: dihydroxyacetone phosphate, PEP: phosphoenolpyruvate.

**Figure 2 jof-08-00822-f002:**
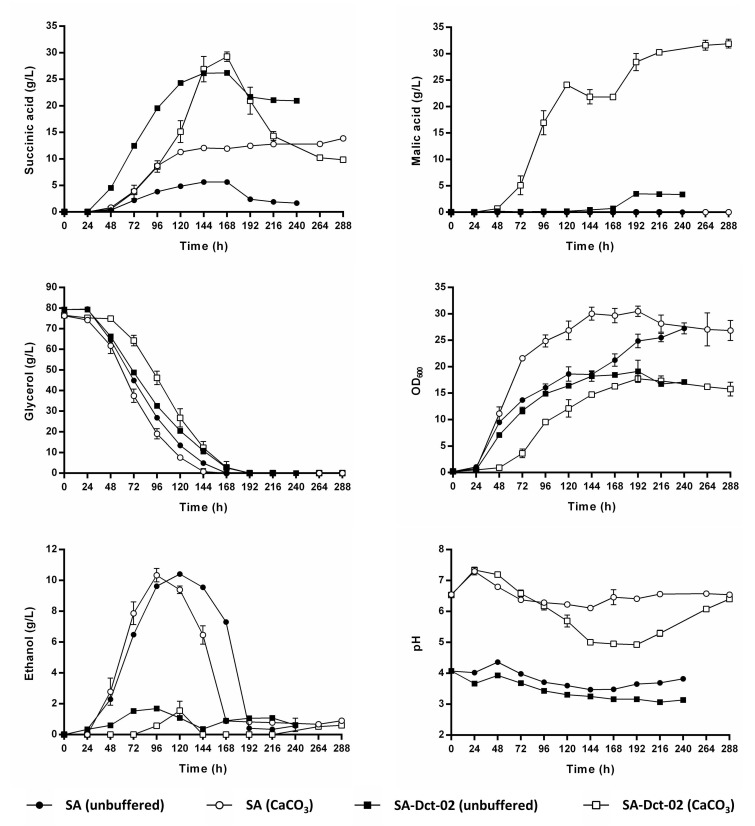
*Saccharomyces cerevisiae* strains SA (● and ○) and SA-Dct-02 (■ and □) cultivated in unbuffered (● and ■) and CaCO_3_-buffered (○ and □) synthetic glycerol media (see composition in materials and methods). The cultivations were performed in 500 mL shake flasks filled with 100 mL medium using urea as the nitrogen source. The unbuffered medium had an initial pH of 4.0. The buffered medium had an initial pH of 6.0 prior to the addition of 30 g/L of CaCO_3_. An HPLC analysis was used to determine the concentrations of succinic acid, malic acid, glycerol, and ethanol in the culture supernatants. Biomass accumulation was measured by optical density at 600 nm (OD_600_). Mean values and standard deviations were determined from three biological replicates.

**Figure 3 jof-08-00822-f003:**
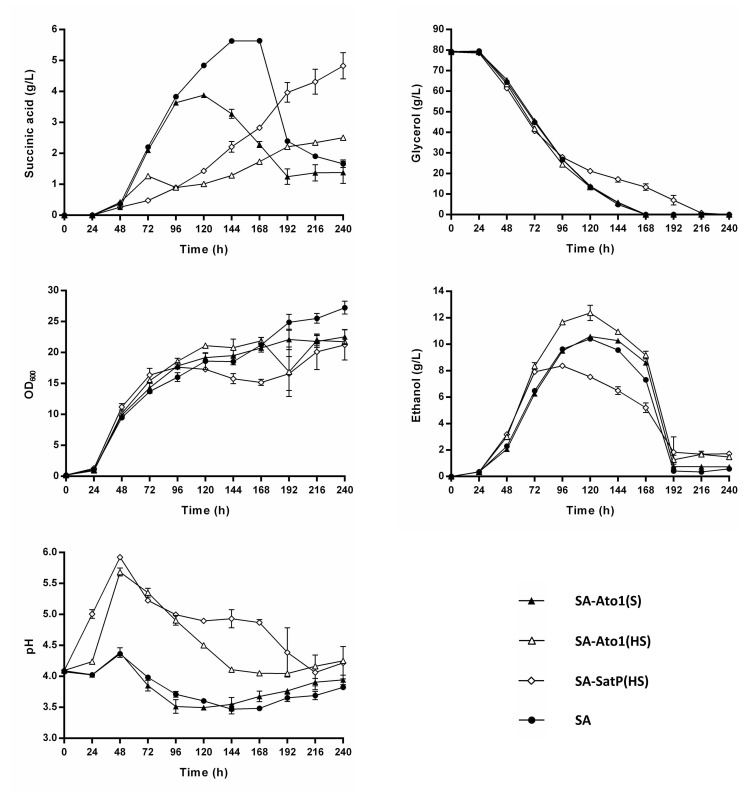
*Saccharomyces cerevisiae* strains SA-Ato1 (S) (▲), SA-Ato1 (HS) (△), SA-SatP (HS) (◇), and SA (●) cultivated in unbuffered synthetic glycerol medium using urea as the nitrogen source (see composition in materials and methods). The cultivations were performed in 500 mL shake flasks filled with 100 mL medium at an initial pH of 4.0. An HPLC analysis was used to determine the concentrations of succinic acid, glycerol, and ethanol in the culture supernatant. Biomass accumulation was measured by optical density at 600 nm (OD_600_). Mean values and standard deviations were determined from three biological replicates.

**Figure 4 jof-08-00822-f004:**
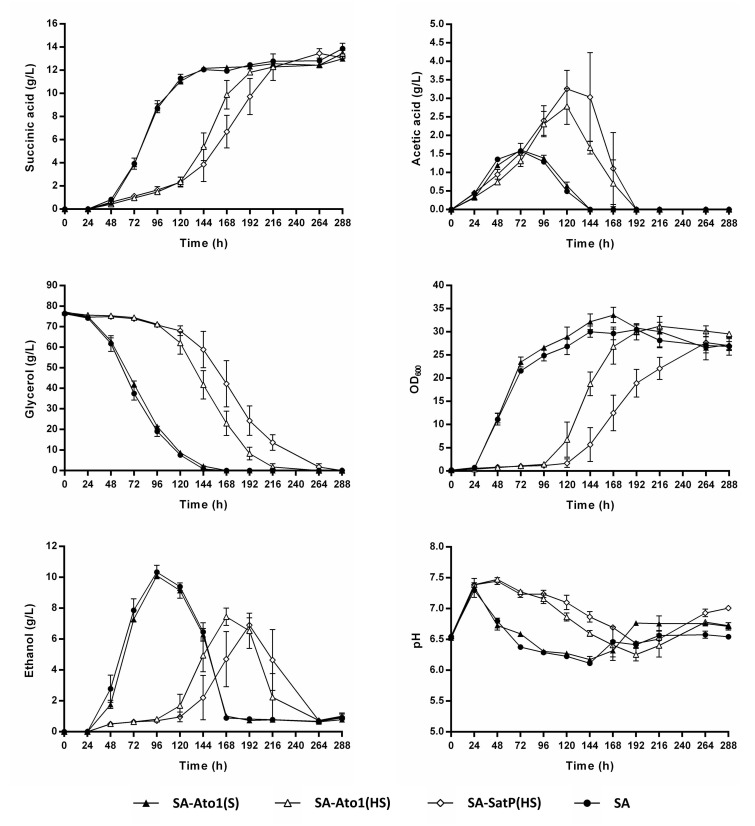
*Saccharomyces cerevisiae* strains SA-Ato1 (S) (▲), SA-Ato1 (HS) (△), SA-SatP (HS) (◇), and SA (●) cultivated in synthetic glycerol medium using urea as the nitrogen source and buffered with 30 g/L of CaCO_3_ (see composition in materials and methods). The cultivations were performed in 500 mL shake flasks filled with 100 mL medium. The initial pH of the medium was 6.0, prior to CaCO_3_ addition. HPLC analysis was used to determine the concentrations of succinic acid, acetic acid, glycerol, and ethanol in the culture supernatant. Biomass accumulation was measured by optical density at 600 nm (OD_600_). Mean values and standard deviations were determined from three biological replicates.

**Figure 5 jof-08-00822-f005:**
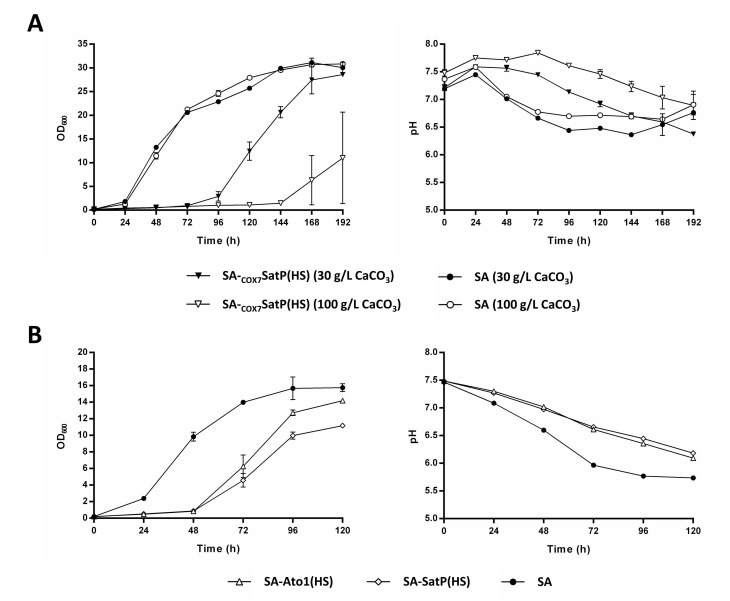
The prolonged lag phase in CaCO_3_-containing medium of the strains which expressed Ato1^E144A, L219A^ and SatP^L131A^ seems to be a consequence of the increased pH. (**A**) Growth of the strains SA-_COX7_SatP (HS) (▼ and ▽) and SA (● and ○) in synthetic glycerol medium using urea as the nitrogen source and buffered with two different concentrations of CaCO_3_: 30 g/L (▼ and ●) and 100 g/L (▽ and ○) (see composition in materials and methods). The initial pH of the medium prior to CaCO_3_ addition was set to 7.0. (**B**) Growth of the strains SA-Ato1 (HS) (△), SA-SatP (HS) (◇), and SA (●) in synthetic glycerol medium using urea as the nitrogen source and buffered with 100 mM potassium phosphate (see composition in materials and methods). The initial pH was set to 7.5. All cultivations (**A**,**B**) were performed in 500 mL shake flasks filled with 100 mL medium. Biomass accumulation was measured by optical density at 600 nm (OD_600_). Mean values and mean absolute deviations were determined from two biological replicates.

**Figure 6 jof-08-00822-f006:**
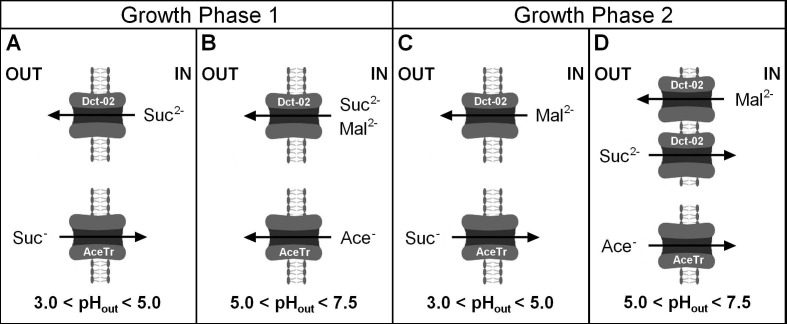
Schematic representation of transport reactions mediated by AceTr homologues and Dct-02 during two distinct cultivation conditions. Growth phase 1 corresponds to glycerol consumption and growth phase 2 corresponds to the utilization of other carbon sources, upon glycerol exhaustion. (**A**) During the phase of glycerol consumption in unbuffered medium, Dct-02 mediated the export, whereas AceTr homologues mediated the import of succinate anions. (**B**) During the phase of glycerol consumption in CaCO_3_-buffered medium, Dct-02 mediated the export of succinate and malate anions, whereas AceTr homologues did not mediate the import or export of succinate anions. In these conditions, Ato1^E144A, L219A^ and SatP ^L131A^ mediated the export of acetate anions. (**C**) After glycerol exhaustion in the unbuffered medium, Dct-02 mediated the export of malate anions. AceTr homologues mediated the import of succinate anions. (**D**) After glycerol exhaustion in a CaCO_3_-buffered medium, Dct-02 mediated the import of succinate and the export of malate anions, whereas AceTr homologues mediated the import of acetate anions. OUT: extracellular; IN: intracellular; Suc: succinic acid; Mal: malic acid; Ace: acetic acid.

**Table 1 jof-08-00822-t001:** Strains used in this study.

*S. cerevisiae* Strain	Relevant Genotype	Reference
CEN 2PW	*ubr2::UBR2_CBS_**_6412-13A_*; *gut1::GUT1_JL1_; YGLC**τ**3::P_TEF1_-Opgdh-T_CYC1_-P_ADH2_-ScDAK1-T_TPS1_-P_TDH3_-CjFPS1-T_RPL15A_*	[23]
SA	*ubr2::UBR2_CBS_* * _6412-13A_ * *; gut1::GUT1_JL1_; YGLC* *τ* *3::P_TEF1_-Opgdh-T_CYC1_-P_ADH2_-ScDAK1-T_TPS1_-P_TDH3_-CjFPS1-T_RPL15A_; YPRCτ3(21)::P_JEN1_-ScMDH3-R-T_IDP1_-P_HOR7_-RofumR-T_DIT1_-P_FBA1_-TbFRD-R-T_ADH1_*	This study
SA-Dct-02	*ubr2::UBR2_CBS_* * _6412-13A_ * *; gut1::GUT1_JL1_; YGLC* *τ* *3::P_TEF1_-Opgdh-T_CYC1_-P_ADH2_-ScDAK1-T_TPS1_-P_TDH3_-CjFPS1-T_RPL15A_; YPRCτ3(21)::P_JEN1_-ScMDH3-R-T_IDP1_-P_HOR7_-RofumR-T_DIT1_-P_FBA1_-TbFRD-R-T_ADH1_; XI-3::P_TDH3_-AnDCT-02-T_CYC1_*	This study
SA-Ato1(S)	*ubr2::UBR2_CBS_* * _6412-13A_ * *; gut1::GUT1_JL1_; YGLC* *τ* *3::P_TEF1_-Opgdh-T_CYC1_-P_ADH2_-ScDAK1-T_TPS1_-P_TDH3_-CjFPS1-T_RPL15A_; YPRCτ3(21)::P_JEN1_-ScMDH3-R-T_IDP1_-P_HOR7_-RofumR-T_DIT1_-P_FBA1_-TbFRD-R-T_ADH1_; XI-3::P_TDH3_-ScATO1^L219A^-T_CYC1_*	This study
SA-Ato1(HS)	*ubr2::UBR2_CBS_* * _6412-13A_ * *; gut1::GUT1_JL1_; YGLC* *τ* *3::P_TEF1_-Opgdh-T_CYC1_-P_ADH2_-ScDAK1-T_TPS1_-P_TDH3_-CjFPS1-T_RPL15A_; YPRCτ3(21)::P_JEN1_-ScMDH3-R-T_IDP1_-P_HOR7_-RofumR-T_DIT1_-P_FBA1_-TbFRD-R-T_ADH1_; XI-3::P_TDH3_-ScATO1^E144A, L219A^-T_CYC1_*	This study
SA-SatP(HS)	*ubr2::UBR2_CBS_* * _6412-13A_ * *; gut1::GUT1_JL1_; YGLC* *τ* *3::P_TEF1_-Opgdh-T_CYC1_-P_ADH2_-ScDAK1-T_TPS1_-P_TDH3_-CjFPS1-T_RPL15A_; YPRCτ3(21)::P_JEN1_-ScMDH3-R-T_IDP1_-P_HOR7_-RofumR-T_DIT1_-P_FBA1_-TbFRD-R-T_ADH1_; XI-3::P_TDH3_-EcSATP^L131A^-T_CYC1_*	This study
SA-_COX7_SatP(HS)	*ubr2::UBR2_CBS_* * _6412-13A_ * *; gut1::GUT1_JL1_; YGLC* *τ* *3::P_TEF1_-Opgdh-T_CYC1_-P_ADH2_-ScDAK1-T_TPS1_-P_TDH3_-CjFPS1-T_RPL15A_; YPRCτ3(21)::P_JEN1_-ScMDH3-R-T_IDP1_-P_HOR7_-RofumR-T_DIT1_-P_FBA1_-TbFRD-R-T_ADH1_; XI-3::P_COX7_-EcSATP^L131A^-T_CYC1_*	This study

**Table 2 jof-08-00822-t002:** Plasmids used in this study.

Plasmid Name	Description	Reference
pUC18-MDH3	Template for the amplification of P*_JEN1_*-*ScMDH3-R*-T*_IDP1_* expression cassette.	[12]
pUC18-RoFUM	Template for the amplification of P*_HOR7_*-*RofumR*-T*_DIT1_* expression cassette.	[12]
pUC18-TbFRD	Template for the amplification of P*_FBA1_-TbFRD-R-*T*_ADH1_* expression cassette.	[12]
pUC18-AnDCT-02 w/o STOP	Codon optimized coding sequence for the dicarboxylic acid transporter Dct-02 from *A. niger*.	[12]
pL219A	Coding sequence for the carboxylic acid transporter variant Ato1^L219A^ from *S. cerevisiae*	[16]
pE144A/L219A	Coding sequence for the carboxylic acid transporter variant Ato1^E144A, L219A^ from *S. cerevisiae*	[16]
pSatP-L131A	Codon optimized coding sequence for the carboxylic acid transporter variant SatP^L131A^ from *E. coli*.	[16]
p414-TEF1p-Cas9-CYC1t-nat1	*CEN6/ARSH4*, *natMX4*, *TEF1p-cas9-CYC1t*	[25]
p414-TEF1p-Cas9-CYC1t-hphMX	*CEN6/ARSH4*, *hphMX*, *TEF1p-cas9-CYC1t*	[12]
p426-SNR52p-gRNA.YPRCτ3-SUP4t-hphMX	2 µm, *hphMX*, *SNR52p-gRNA*.*YPRCτ3-SUP4t*	[11]
pCfB3045	2 µm, *natMX6*, *SNR52p-gRNA*.XI-3-*SUP4t*	[26]

## Data Availability

Not applicable.

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
