# Peer review of "The Dicarboxylate Transporters from the AceTr Family and Dct-02 Oppositely Affect Succinic Acid Production in S. cerevisiae"

_jof, 2022, doi:10.3390/jof8080822_

Round 1
Reviewer 1 Report
In this manuscript the authors report that the dicarboxylate anion channels from 2 different families, the Acetate Uptake Transporters (AceTr) and the SLAC1 family, Dct-02 Aspergillus niger transporter, have different impact on extracellular succinic acid accumulation in the yeast Saccharomyces cerevisiae cell factory used. In particular, in the yeast strain they have metabolically engineering to overproduce succinic acid from glycerol and CO2, they express transporter alleles from the AceTr family (Ato1L219A, Ato1E144A,L219A) and its bacterial homologue SatPL131A, and compare them in regard mainly to succinic acid export with an isogenic strain expressing the Dct-2 transporter. Succinic acid export is a process essential for efficient and cost-effective succinic acid production from microorganisms. The tested AceTr alleles, opposite to Dct-02 that prefers divalent succinate, hinder effective succinic acid production in media with low pH, since they seem to rather act as succinic acid importers at low external pH ie external pH modulates their activity, and prefer monovalent succinate. Moreover, the authors provide evidence that Dct-02 also imports succinic acid and exports malate, which is an unwanted by-product in respect to industrial succinic acid production via yeast cell factories and thus it has to be engineered.
General comments:
To my opinion this is a well-written manuscript, where the results are clearly represented and the conclusions made are supported by the results. It is a manuscript acceptable in its present form.
Author Response
Manuscript ID: jof-1811920
The dicarboxylate transporters from the AceTr family and Dct-02 oppositely affect succinic acid production in S. cerevisiae
We acknowledge the reviewers for the critical reading of the manuscript as well as their valuable suggestions, which contributed to improving the overall quality of the manuscript. We followed point-by-point all the proposed comments and suggestions and outlined every change made, as described below. In addition, we highlighted in the manuscript the alterations of the revised version.
Reviewer comments:
Reviewer #1
In this manuscript the authors report that the dicarboxylate anion channels from 2 different families, the Acetate Uptake Transporters (AceTr) and the SLAC1 family, Dct-02 Aspergillus niger transporter, have different impact on extracellular succinic acid accumulation in the yeast Saccharomyces cerevisiae cell factory used. In particular, in the yeast strain they have metabolically engineering to overproduce succinic acid from glycerol and CO2, they express transporter alleles from the AceTr family (Ato1L219A, Ato1E144A,L219A) and its bacterial homologue SatPL131A, and compare them in regard mainly to succinic acid export with an isogenic strain expressing the Dct-02 transporter. Succinic acid export is a process essential for efficient and cost-effective succinic acid production from microorganisms. The tested AceTr alleles, opposite to Dct-02 that prefers divalent succinate, hinder effective succinic acid production in media with low pH, since they seem to rather act as succinic acid importers at low external pH ie external pH modulates their activity, and prefer monovalent succinate. Moreover, the authors provide evidence that Dct-02 also imports succinic acid and exports malate, which is an unwanted by-product in respect to industrial succinic acid production via yeast cell factories and thus it has to be engineered.
General comments:
To my opinion this is a well-written manuscript, where the results are clearly represented and the conclusions made are supported by the results. It is a manuscript acceptable in its present form.
Reply: The reviewer did not suggest any additional modifications.
Reviewer 2 Report
It will be useful to add a brief conclusion section.
Author Response
Manuscript ID: jof-1811920
The dicarboxylate transporters from the AceTr family and Dct-02 oppositely affect succinic acid production in S. cerevisiae
We acknowledge the reviewers for the critical reading of the manuscript as well as their valuable suggestions, which contributed to improving the overall quality of the manuscript. We followed point-by-point all the proposed comments and suggestions and outlined every change made, as described below. In addition, we highlighted in the manuscript the alterations of the revised version.
Reviewer comments:
Reviewer #2
It will be useful to add a brief conclusion section.
Reply: We added a conclusion section in which we state the most important accomplishments and observations of our study. By combining the suggestions from Reviewer #2 and Reviewer #3, we also included a new figure (Fig. 6) as a schematic representation of behaviour of Dct-02 and AceTr transporters in the conditions here studied.
Reviewer 3 Report
In the manuscript, “The dicarboxylate transporters from the AceTr family and Dct- 2
02 oppositely affect succinic acid production in S. cerevisiae” the authors performed the set of experiments, suggesting that the AceTr channels prefer monovalent succinate whereas Dct-02 prefers divalent succinate anions. Moreover, the authors showed the interesting feature of the Dct-2 which moves succinate bidirectionally across the membrane. Overall, I think this is a comprehensive and very informative study that would be a great addition to the Journal of Fungi.
I have a couple of suggestions that I think would make the manuscript easier to follow and questions for the authors to address.
1. A schematic of the process of succinate and malate translocation facilitated by channels (Dct-02, AceTr, etc.) would be very helpful for the reader. For example, a drawing like Fig.1 shows the plasma membrane, the position of channels, and the path of succinate and/or malate translocation under different growing conditions.
2. Can the authors quantify the protein expression level of Dct-02 and AceTr channels? Did they check the channel(s) expression level in different growing stages? Would the different media (buffered vs unbuffered) affect protein expression?
3. In Line 343, what is the pH value of the later phase of buffered cultivation? This would help to determine the charge of succinate which has pKa of 4.2 and 5.6. Did they measure the pH at this growing phase?
4. To claim Dct-02 as an importer, the authors should consider a transport assay1,2 that adds succinate in buffer and measures the accumulation of succinate inside the cell. A time course of 1 to 20 minutes usually is sufficient for this experiment. This could help eliminate some environmental factors during fermentation which could complicate the explanation of the results.
5. The activity of SLAC1 channels is regulated through phosphorylation3. It is interesting to see if this could be a strategy for further engineering. Can authors comment on this?
Reference
1. Sauer DB, Song J, Wang B, Hilton JK, Karpowich NK, Mindell JA, Rice WJ, Wang DN. Structure and inhibition mechanism of the human citrate transporter NaCT. Nature. 2021 Mar;591(7848):157-161. doi: 10.1038/s41586-021-03230-x. Epub 2021 Feb 17. PMID: 33597751; PMCID: PMC7933130.
2. Su CC, Bolla JR, Kumar N, Radhakrishnan A, Long F, Delmar JA, Chou TH, Rajashankar KR, Shafer WM, Yu EW. Structure and function of Neisseria gonorrhoeae MtrF illuminates a class of antimetabolite efflux pumps. Cell Rep. 2015 Apr 7;11(1):61-70. doi: 10.1016/j.celrep.2015.03.003. Epub 2015 Mar 26. PMID: 25818299; PMCID: PMC4410016.
3. Deng YN, Kashtoh H, Wang Q, Zhen GX, Li QY, Tang LH, Gao HL, Zhang CR, Qin L, Su M, Li F, Huang XH, Wang YC, Xie Q, Clarke OB, Hendrickson WA, Chen YH. Structure and activity of SLAC1 channels for stomatal signaling in leaves. Proc Natl Acad Sci U S A. 2021 May 4;118(18):e2015151118. doi: 10.1073/pnas.2015151118. PMID: 33926963; PMCID: PMC8106318.
Author Response
Manuscript ID: jof-1811920
The dicarboxylate transporters from the AceTr family and Dct-02 oppositely affect succinic acid production in S. cerevisiae
We acknowledge the reviewers for the critical reading of the manuscript as well as their valuable suggestions, which contributed to improving the overall quality of the manuscript. We followed point-by-point all the proposed comments and suggestions and outlined every change made, as described below. In addition, we highlighted in the manuscript the alterations of the revised version.
Reviewer comments:
Reviewer #3
In the manuscript, “The dicarboxylate transporters from the AceTr family and Dct-02 oppositely affect succinic acid production in S. cerevisiae” the authors performed the set of experiments, suggesting that the AceTr channels prefer monovalent succinate whereas Dct-02 prefers divalent succinate anions. Moreover, the authors showed the interesting feature of the Dct-2 which moves succinate bidirectionally across the membrane. Overall, I think this is a comprehensive and very informative study that would be a great addition to the Journal of Fungi.
I have a couple of suggestions that I think would make the manuscript easier to follow and questions for the authors to address.
- A schematic of the process of succinate and malate translocation facilitated by channels (Dct-02, AceTr, etc.) would be very helpful for the reader. For example, a drawing like Fig.1 shows the plasma membrane, the position of channels, and the path of succinate and/or malate translocation under different growing conditions.
Reply: We added a new figure (Fig. 6) which showcases the observed transport of organic acids by the respective channels, based on 2 cultivation conditions and 2 stages of cultivation (before and after glycerol exhaustion). We combined this figure with the suggestion of Reviewer #2 to include the Conclusions section.
- Can the authors quantify the protein expression level of Dct-02 and AceTr channels? Did they check the channel(s) expression level in different growing stages? Would the different media (buffered vs unbuffered) affect protein expression?
Reply: We did not plan to quantify the protein expression levels of Dct-02 and AceTr channels in this study. Thus, we did not check the expression levels of the respective channels in different phases of growth. To accomplish such work, we would have to construct a new set of strains with appropriate fusion tags to be detected by Western blot. In addition, we would have to check if the activity of these transporters is not affected by the fusion tags, thus having to repeat most of the fermentation experiments. Better understanding of regulation and post-translational modification of Dct-02 and AceTr channels would certainly be valuable for the fundamental knowledge about these channels while also potentially bringing new ideas for their further engineering, however, these research questions were not in the scope of our current work.
To our knowledge, there are no studies which demonstrated the regulation of Dct-02 and AceTr channels on the level of their protein expression, based on the pH value of the medium or growth phase of yeast cells.
- In Line 343, what is the pH value of the later phase of buffered cultivation? This would help to determine the charge of succinate which has pKa of 4.2 and 5.6. Did they measure the pH at this growing phase?
Reply: The pH values during the respective cultivation phase were already available in Figure 2, but we have now included them into the main text (Line 346) for easier data interpretation. The pH values gradually rose from 5.0 to 6.4 during the succinic acid reconsumption phase.
- To claim Dct-02 as an importer, the authors should consider a transport assay1,2 that adds succinate in buffer and measures the accumulation of succinate inside the cell. A time course of 1 to 20 minutes usually is sufficient for this experiment. This could help eliminate some environmental factors during fermentation which could complicate the explanation of the results.
Reply: We have performed radioactively labelled succinate transport assays in S. cerevisiae cells of different strain background than the one utilized in this study, which confirmed the succinic acid import activity of Dct-02. To accomplish the reviewer request, we would have to present significant amount of work which is beyond the objective of the current manuscript. The full characterization of Dct-02 kinetics, energetics, and specificity will be presented in another manuscript. For the above-mentioned reasons, we prefer to not disclose radioactive uptakes in this study.
- The activity of SLAC1 channels is regulated through phosphorylation3. It is interesting to see if this could be a strategy for further engineering. Can authors comment on this?
Reply: The authors of the respective publication found 13 phosphorylation sites in the N-terminus and 1 in the C-terminus of the SLAC1 channel from Arabidopsis thaliana, which is a distant homologue of Dct-02. Some of these sites were shown to play a major role in the activity of the AtSLAC1 channel i.e., phosphorylation of these sites promoted the activity of AtSLAC1. By performing phosphomimetics (swapping serine and/or threonine residues with aspartate residues), authors mimicked the state of constant phosphorylation, thus improving the activity of AtSLAC1. Since Dct-02 is a distant homologue of AtSLAC1, one could hypothesize that Dct-02 may also possess such mode of regulation. According to our analysis, the termini of Dct-02 and AtSLAC1 do not show a high degree of homology. Moreover, the termini of Dct-02 are much shorter. Nonetheless, a dozen of serine and threonine residues can be found in the termini of Dct-02, and some of them appear to be homologous to the ones of AtSLAC1. If the hypothesis above is true, one could engineer an improved activity of Dct-02 by performing phosphomimetics within its serine and threonine residues. Thus, we decided to include the idea for Dct-02 engineering in the discussion (Line 531) and cite the work about AtSLAC13: “Recent findings about activation of Dct-02 homologue AtSLAC1 from Arabidopsis thaliana via posttranslational modification [37] suggest that the export activity of Dct-02 could be improved by introducing phosphomimetic substitutions in its putative phosphorylation sites.”
Reference:
- Sauer DB, Song J, Wang B, Hilton JK, Karpowich NK, Mindell JA, Rice WJ, Wang DN. Structure and inhibition mechanism of the human citrate transporter NaCT. Nature. 2021 Mar;591(7848):157-161. doi: 10.1038/s41586-021-03230-x. Epub 2021 Feb 17. PMID: 33597751; PMCID: PMC7933130.
- Su CC, Bolla JR, Kumar N, Radhakrishnan A, Long F, Delmar JA, Chou TH, Rajashankar KR, Shafer WM, Yu EW. Structure and function of Neisseria gonorrhoeae MtrF illuminates a class of antimetabolite efflux pumps. Cell Rep. 2015 Apr 7;11(1):61-70. doi: 10.1016/j.celrep.2015.03.003. Epub 2015 Mar 26. PMID: 25818299; PMCID: PMC4410016.
- Deng YN, Kashtoh H, Wang Q, Zhen GX, Li QY, Tang LH, Gao HL, Zhang CR, Qin L, Su M, Li F, Huang XH, Wang YC, Xie Q, Clarke OB, Hendrickson WA, Chen YH. Structure and activity of SLAC1 channels for stomatal signaling in leaves. Proc Natl Acad Sci U S A. 2021 May 4;118(18):e2015151118. doi: 10.1073/pnas.2015151118. PMID: 33926963; PMCID: PMC8106318.